# Post-Stroke Infections: Insights from Big Data Using Clinical Data Warehouse (CDW)

**DOI:** 10.3390/antibiotics12040740

**Published:** 2023-04-12

**Authors:** Moa Jung, Hae-Yeon Park, Geun-Young Park, Jong In Lee, Youngkook Kim, Yeo Hyung Kim, Seong Hoon Lim, Yeun Jie Yoo, Sun Im

**Affiliations:** 1Department of Rehabilitation Medicine, Bucheon St. Mary’s Hospital, College of Medicine, The Catholic University of Korea, Seoul 06591, Republic of Korea; moa1553@naver.com (M.J.);; 2Department of Rehabilitation Medicine, Seoul St. Mary’s Hospital, College of Medicine, The Catholic University of Korea, Seoul 06591, Republic of Korea; 3Department of Rehabilitation Medicine, Yeouido St. Mary’s Hospital, College of Medicine, The Catholic University of Korea, Seoul 06591, Republic of Korea; 4Department of Rehabilitation Medicine, Uijeongbu St. Mary’s Hospital, College of Medicine, The Catholic University of Korea, Seoul 06591, Republic of Korea; 5Department of Rehabilitation Medicine, St. Vincent’s Hospital, College of Medicine, The Catholic University of Korea, Seoul 06591, Republic of Korea

**Keywords:** risk factors, functional level (modified Barthel), infection, big data, stroke, steroids, pneumonia, urinary tract infection, electronic health record

## Abstract

This study analyzed a digitized database of electronic medical records (EMRs) to identify risk factors for post-stroke infections. The sample included 41,236 patients hospitalized with a first stroke diagnosis (ICD-10 codes I60, I61, I63, and I64) between January 2011 and December 2020. Logistic regression analysis was performed to examine the effect of clinical variables on post-stroke infection. Multivariable analysis revealed that post-stroke infection was associated with the male sex (odds ratio [OR]: 1.79; 95% confidence interval [CI]: 1.49–2.15), brain surgery (OR: 7.89; 95% CI: 6.27–9.92), mechanical ventilation (OR: 18.26; 95% CI: 8.49–44.32), enteral tube feeding (OR: 3.65; 95% CI: 2.98–4.47), and functional activity level (modified Barthel index: OR: 0.98; 95% CI: 0.98–0.98). In addition, exposure to steroids (OR: 2.22; 95% CI: 1.60–3.06) and acid-suppressant drugs (OR: 1.44; 95% CI: 1.15–1.81) increased the risk of infection. On the basis of the findings from this multicenter study, it is crucial to carefully evaluate the balance between the potential benefits of acid-suppressant drugs or corticosteroids and the increased risk of infection in patients at high risk for post-stroke infection.

## 1. Introduction

Infections are a frequent complication in stroke patients due to their increased susceptibility [1]. Stroke causes the activation of immune feedback loops between the brain and peripheral organs [2], inducing immunosuppression with characteristics of lymphocytopenia and T-cell suppression [3,4]. This stroke-induced immunosuppression might cause an increased infection rate after acute stroke. Although previous studies have shown variable rates of infection, in a recent meta-analysis, the overall pooled infection rate was 30% of acute stroke patients [5].

Post-stroke infections are unique from other infections in elderly or critically ill patients, in that the central nervous system injury is hypothesized to lead to induced immunosuppression [1,6]. They have been shown to have a negative impact on outcomes for acute stroke patients, including increased mortality and poor clinical outcomes [7,8]. A study showed that, in patients with a hospital length of stay greater than 7 days, pneumonia accounted for 12.2% of in-hospital deaths and 6.4% of poor outcomes (modified Rankin scale score ≥ 3) at discharge [7]. Additionally, another study showed that infections can result in poor outcomes 1 year after admission [8]. In the setting of post-stroke infections, immune responses in the blood and in the brain are implicated in longer-term cognitive decline and dementia after stroke [9]. Therefore, understanding and identifying the risk factors regarding post-stroke infections is a critical issue in the clinical management of acute stroke patients.

Previous studies on the predictors of infection in stroke patients revealed that nasogastric tube use, atrial fibrillation, total anterior circulation syndrome, and urinary catheter usage increased the risk of post-stroke infection [10]. Another meta-analysis on risk factors for post-stroke pneumonia showed that advanced age, male sex, stroke severity, dysphagia, nasogastric tube use, diabetes, mechanical ventilation, smoking, chronic obstructive pulmonary disease, and atrial fibrillation are predictors [11,12,13]. However, most studies focused on pneumonia, and research on all-cause post-stroke infections is limited. To address this, studies on all-cause post-stroke infections are needed, as urinary tract infections in stroke patients are just as common as pneumonia, and both increase the risk of unfavorable outcomes [5].

In addition to clinical variables, medications are modifiable factors that can increase the risk of infection in the general population. In recent years, clinical data warehouses have become an important resource for healthcare research because they centralize and integrate large amounts of clinical data from various sources, allowing for efficient storage, retrieval, and analysis of patient information. This can improve decision making and patient outcomes by enabling a better understanding of disease processes and by providing access to the patient’s complete medical history, including test results, medication history, and treatment outcomes, thus allowing for more accurate predictions. Including medication and laboratory data from a clinical data warehouse (CDW) analysis can facilitate examination of the relationship between medication use and post-stroke infections, allowing the development of preventive strategies and drug safety procedures in relation to infections in stroke patients [14].

Therefore, the aim of this study was to evaluate the association between clinical factors, including medication and lab tests, and the incidence of all-cause post-stroke infections during hospitalization using electronic health records from multiple university-affiliated stroke centers. The collected data are analyzed to examine the effect of factors such as medical comorbidities, stroke treatments, medication use, and enteral feeding on post-stroke infections, as well as provide a model that may help predict post-stroke infections in an accurate manner.

## 2. Results

### 2.1. Baseline Characteristics

From January 2011 to December 2020, 41,236 patients aged 18 years or older were diagnosed with stroke. Of these, 24,250 patients fulfilled the criteria of hospitalization at the time of diagnosis, receiving treatment in a neurology, neurosurgery, or rehabilitation department, hospitalization for a duration exceeding 48 h, and absence of infection within 2 days of admission. The final sample for analysis comprised 6518 patients after excluding individuals with missing data of interest. A flowchart of the patient selection process is illustrated in Figure 1.

Table 1 displays the baseline characteristics of the study population, which consisted of 6518 patients. The median time between stroke onset and subsequent infection was 9 days (interquartile range of 5–14 days). Moreover, the average period between brain surgery and infection was 11.09 days. The prevalence of infection among patients was found to be 12.2%, with 797 of 6518 patients diagnosed with infection (Figure 2).

The results indicated that, compared to the group without infection, patients in the infection group had a higher prevalence of male sex, prior brain surgery, use of mechanical ventilation, enteral tube feeding, and exposure to various medications, including antipsychotics, opioids, dopamine agonists, serotonin-related antidepressants, steroids, and acid-suppressant drugs. Additionally, these patients had higher Charlson comorbidity index (CCI) scores. On the other hand, patients without infection were found to have a higher exposure to antiplatelet medications (such as aspirin and clopidogrel) and higher modified Barthel index (MBI) during their hospital admissions. In contrast, the group of patients with atrial fibrillation exhibited lower rates of infection.

### 2.2. Clinical Variables Associated with Post-Stroke Infection

A logistic regression analysis was conducted to evaluate the relationship between clinical variables and post-stroke infection. The results of the univariable and multivariable analyses are presented in Table 2. The multivariable analysis showed that post-stroke infection was significantly associated with several clinical variables, including male sex (odds ratio [OR] 1.79; 95% confidence interval [CI] 1.49–2.16), hospital length of stay (OR 1.001; 95% CI 1.000–1.002), brain surgery intervention for stroke (OR 7.84; 95% CI 6.24–9.87), use of mechanical ventilation (OR 18.11; 95% CI 8.40–44.04), enteral tube feeding (OR 3.61; 95% CI 2.95–4.43), MBI score (OR 0.98; 95% CI 0.98–0.98), atrial fibrillation (OR 0.62; 95% CI 0.40–0.93), exposure to steroids (OR 2.18; 95% CI 1.57–3.01), and acid-suppressant drugs (OR 1.43; 95% CI 1.15–1.80) (Table 2). The receiver operating characteristic (ROC) curve analysis revealed an area under the curve of 0.86 (95% CI 0.85–0.88) with a sensitivity of 82.7% and a specificity of 77.3% (Figure 3).

## 3. Discussion

The results of our study showed that male sex, a history of brain surgery (OR 7.84, 95% CI 6.24–9.87), use of mechanical ventilation (OR 18.11, 95% CI 8.40–44.04), and enteral tube feeding (OR 3.61, 95% CI 2.95–4.43) were significantly associated with a higher risk of infection in acute stroke patients. The use of steroids and acid-suppressant agents was also found to increase the risk of infection. Conversely, exposure to antipsychotics and benzodiazepines, which have previously been implicated in infection due to their sedative and aspiration-inducing effects, did not result in an increased odds ratio. A higher functional status, as indicated by high MBI, was found to be a protective factor against infection. While the length of hospital stay showed statistically significant values (OR 1.001, 95% CI 1.000–1.002) the small difference between the upper and lower bounds of the confidence interval may not be clinically or practically significant. Overall, the findings from our retrospective analysis of multiple centers provide valuable insights into the risk factors that may increase the likelihood of post-stroke infections.

This study, which analyzed electronic health data from five university-affiliated centers, revealed a significant association between male sex, brain surgery, mechanical ventilation, and enteral tube feeding and the risk of post-stroke infection, which is consistent with previous research [10,11,15,16,17,18,19,20,21]. In neurosurgical patients, postoperative pneumonia incidence is known to be 0.6–27%, according to the different types of diseases and procedures [22]. Post-brain surgery patients are at a higher risk of developing infections due to several factors, such as the use of invasive medical procedures, including intubation, ventilation, and the placement of medical devices such as shunts, drains, and catheters [20,21,23]. Additionally, the prolonged hospitalization required after surgery further increases the risk of infection [24]. Moreover, brain surgery is usually performed for patients with life-threatening conditions, who often have more severe brain injury and greater neurological impairments [25], which may further increase their risk of infection.

One of the interesting findings from our study is that higher functional status, as reflected by higher MBI scores, is significantly associated with reduced odds of post-stroke infection, even after controlling for severe medical and neurological conditions such as brain surgery and mechanical ventilation. Early rehabilitation or mobilization, which improves patients’ functional abilities, may play a critical role in reducing infectious complications and improving neurological outcomes [26,27]. These findings are supported by previous studies and best clinical practice guidelines for stroke rehabilitation [28], which recommend referral for functional assessment and therapy within 48 h of admission. Although the primary objectives of this study were not focused on assessing the preventive effects of early rehabilitation in the ICU, these findings are nonetheless clinically relevant. Previous research has shown that early mobilization can provide significant benefits for critically ill patients, including a reduction in the number of days on mechanical ventilation and, thus, a potential decrease in the incidence of complications related to ventilator-associated pneumonia [29,30].

A unique aspect of our study was the analysis of big data from electronic health records, which enabled us to examine the effect of medication exposure in conjunction with other clinical characteristics. For example, our results showed that the use of steroids was significantly associated with an increased risk of post-stroke infection. Systemic corticosteroids may sometimes be used in some severe cases to control for post-stroke brain edema [31,32], but their benefits have been controversial. The increased risk of infection could be due to the immunosuppressive and anti-inflammatory effects of steroids [33]. One may also postulate the possibility of corticosteroid administration to patients with severe septic shock [34], thus leading to higher prevalence and associated risk. However, it should be noted that the cohort included in this study consisted of acute stroke patients undergoing stroke intervention, and those with severe medical conditions with septic shock requiring steroid administration would likely have been admitted to the infection department and, therefore, excluded from our cohort.

The results regarding the use of acid suppressant drugs, such as proton pump inhibitors and histamine-2 receptor antagonists, were also consistent with the results of previous studies on the association of such medications with hospital-acquired pneumonia or post-stroke pneumonia [35,36,37,38]. From some studies, it has been suggested that proton pump inhibitors may inhibit leukocyte function, exacerbating stroke-induced immunosuppression and increasing the risk of infection in stroke patients [39,40,41]; this inhibition could exacerbate stroke-induced immunosuppression and increase the risk of infection in stroke patients. In the context of the intensive care unit or high-risk patients, the use of acid-suppressant drugs to prevent gastrointestinal bleeding must be carefully balanced against the possibility of pneumonia [42] and infection as indicated by our study results.

Antipsychotics and benzodiazepines, which are known to increase the risk of infection by causing sedation and aspiration [43,44], were found to be prescribed at higher rates in the group of patients with infections. However, the multivariable analysis did not demonstrate a significant increase in the odds ratios (ORs) of post-stroke infection with these medications. Further studies using propensity score analysis are needed to confirm the effect of antipsychotics on post-stroke infections. The logistic regression model developed in this study, which utilized electronic health data, showed strong diagnostic properties for post-stroke infection with an area under the receiver operating characteristic curve (AUC-ROC) of 0.86 (95% CI: 0.85–0.88) and high levels of sensitivity. These results are comparable to and slightly better than those of a previously reported predictive model of post-stroke pneumonia (AUC-ROC 0.85; 95% CI: 0.80–0.91) [18]. Our model was obtained from a clinical data warehouse that included all post-stroke infections; it has the potential to identify individuals at high risk of infection with high sensitivity in an automated manner during acute management and may help improve the efficiency and quality of post-stroke care.

Our findings suggest the significance of preventive measures during the early stages of ICU care after brain surgery, particularly during mechanical ventilation and enteral nutrition, in reducing the risk of post-stroke infection. While some measures have been suggested for acute post-stroke management to prevent infection, our study recommends several possible preventive measures. For example, although the use of antibiotics for prophylaxis in stroke patients has shown conflicting results in preventing pneumonia [45,46], with no improvement in functional outcome observed [47], a single dose of antibiotic prophylaxis at intubation has been shown to reduce the incidence of ventilator-associated pneumonia [48]. Adopting a semi-recumbent position (≥30°) and implementing subglottic secretion drainage have also been shown to reduce ventilator-associated pneumonia [49,50]. In brain surgery patients, measures such as perioperative oxygenation, maintaining normal body temperature, and goal-directed fluid management based on continuous stroke volume variation and stroke volume monitoring can help to reduce surgery-related infection [51,52]. The type of enteral tube feeding does not seem to affect pneumonia prevention [53,54]. Although the evidence is limited, routine oral care using chlorhexidine gluconate may offer potential benefits in reducing the risk of pneumonia [55,56].

A major discrepant finding of our study suggesting that atrial fibrillation does not increase the risk of post-stroke infection contradicts previous studies. Further research is still needed to clarify the association between post-stroke infection and atrial fibrillation. Previous studies have indicated that factors such as prolonged immobility, advanced age, and anticoagulant therapy may increase the risk of post-stroke infection in patients with atrial fibrillation. Various studies have suggested a potential association between inflammation and the incidence of atrial fibrillation (AF) [57]. Furthermore, a retrospective study involving administrative data, with a sample size of 49,082 patients with severe sepsis, demonstrated that individuals with new-onset AF during their hospitalization had a greater stroke risk compared to those with preexisting AF (adjusted OR, 3.63 [95% CI, 2.51–5.25]) [58]. These findings may suggest that the infection causing AF may increase the risk of stroke. However, to fully understand this hypothesis, further studies are needed to compare stroke risk in patients with AF caused by infection and in those with AF without any accompanying infection. However, additional studies are required to establish a clearer understanding of the relationship. There is evidence suggesting that aspirin and clopidogrel may have a protective effect against infection [59,60], leading to the hypothesis that anticoagulants may have similar protective effects. Additionally, it is important to further examine the impact of drugs that control ventricular rate in atrial fibrillation, such as beta-blockers or calcium channel blockers, on post-stroke infection. For instance, it has been suggested that propranolol reduces the risk of oropharyngeal dysphagia [61]; this drug has also been demonstrated to prevent suppression of cellular immune responses after experimental stroke, offering a potential mechanism for reducing post-stroke infection risk [62].

In terms of comorbidities, previous studies have included some individual medical problems in the analysis, with a focus mostly on atrial fibrillation and diabetes. Although the combined odds ratio of diabetes and atrial fibrillation was significant in a meta-analysis [11], whether or not they were individually significant differed from study to study [10,15,16]. This is probably because the effect of each diseases differs depending on the overall conditions such as the patient’s age or the coexistence of various other chronic diseases. To address these inconsistencies, we incorporated the CCI in this study. This index provides an overall measure of a patient’s comorbid disease status, and it was used to investigate whether the presence of multiple comorbidities is associated with an increased risk of infection.

The results of this study regarding the prevalence of post-stroke infection presented a discrepancy compared to previous studies. Previous studies have reported wide-ranging estimates for the prevalence of post-stroke infection [8], with a meta-analysis of 87 studies reporting a pooled rate of 30% [5]. In this study, the prevalence of infection was 12.2%, which is lower than the rates reported in previous studies. This difference may be due to the strict criteria used to classify infection in this study, which involved stringent cutoff values for laboratory results and a relatively short median follow-up period of 10 days. In addition, the limited availability of MBI scores for all patients may have influenced the results. However, the prevalence of infection in this study was consistent with that observed in patients who dropped out due to missing data, including the Barthel index, with a rate of 11.7%. This consistency suggests that the data from this study are reliable and that the potential for sample selection bias is limited.

Some limitations must be considered when interpreting the results of the study. First, the study was retrospective and cross-sectional, which means that a causal relationship between factors and infection could not be established. The timing of drug exposure and infection is also unclear, which raises the possibility of protopathic bias [63]. Second, while previous hospitalization, particularly within the preceding 90 days, is a known risk factor for infection [64], we did not include data from patients who had previously been hospitalized or had a prior infection. This allowed us to focus on patients with de novo post-stroke infections. However, it is important to note that our study only included medical records within the CDW; therefore, we could not exclude patients who may have had prior hospitalizations at other medical centers, as this information was not accessible through the electronic platform [65]. It is important to note that only certain types of information can be retrieved through CDW. In particular, data contained in free text fields could not be automatically retrieved. For example, the initial neurological status of the patients (e.g., Glasgow coma scale (GCS) score) was unavailable for automatic retrieval in the study due to the unstructured nature of data [66]. Instead, we were able to retrieve the MBI scores, which reflect the functional levels and can provide valuable information on a patient’s functional independence in activities of daily living, which can aid in clinical decision making and help guide rehabilitation planning. By contrast, the GCS may be subject to inconsistent predictive power and may be influenced by sedative use [67]. The study relied solely on laboratory data, prescription information, and diagnostic codes from electronic medical records, limiting the information used in the analysis. In the identification process of post-stroke infection, we used both leukocytosis and the administration of intravenous antibiotics due to the low accuracy of ICD-10 codes in identifying infections. However, early stroke can show leukocytosis without infection, and infections without leukocytosis exist, such as in the elderly with poor bone marrow reserve, immunosuppressed patients with neutropenia, and in cases of severe sepsis with pancytopenia [68,69]. Furthermore, the relationship between brain lesion characteristics, volume, and imaging and the risk of infection has not been fully investigated. Future studies are needed to select the accurate infected population and compare the risk of infection based on specific brain lesion characteristics and genetic information. Furthermore, a Foley catheter provides a direct pathway for bacteria to enter the bladder, increasing the risk of infection. Our dataset in this study did not include information on Foley catheters [70]; however, Foley catheters may not be the sole source of UTI, considering that stroke-induced immunosuppression and increased neurogenic bladder retention may also increase the risk of UTI [71].

Several past studies have shown that post-stroke infections may have an impact on patient outcomes. For instance, pneumonia and urinary tract infections have been found to increase the risk of unfavorable outcomes, with the former even showing an odds ratio of 3.62 for mortality, along with others citing the immunological effects of infections as possible causes leading to poor clinical outcome [5]. While it is crucial to investigate whether specific pathogens are linked to worse outcomes, this remains an area that requires more extensive research. Unfortunately, our analysis was limited by the mixed pathogen results obtained from the electronic database and the fact that the clinical period under observation was limited to the admission period without any follow-up period for analysis.

Lastly, the study was limited in that it did not include information on antithrombotic drugs or anticoagulants commonly prescribed for atrial fibrillation, resulting in inconclusive findings regarding their effect on the risk of post-stroke infection. Other drugs also not included as variables in this study associated with infection risk include immunomodulatory drugs, chemotherapy agents, and inhaled corticosteroids or mucolytics. The latter (e.g., N-acetylcysteine agents) have not shown certain effectiveness in preventing the exacerbation of chronic obstructive pulmonary disease [72,73,74,75], although their use has been linked to improved outcome in stroke models [76,77].

Further research is needed to examine the relationship between these agents and the risk of post-stroke infection. This should include a comprehensive analysis of pharmaceutical agents used to regulate ventricular rate in atrial fibrillation, such as beta blockers, calcium channel blockers, digoxin, amiodarone, and ACE inhibitors, as these have been shown to reduce the risk of post-stroke aspiration pneumonia.

## 4. Materials and Methods

### 4.1. Study Design and Participants

This study was a post hoc study based on medical records retrieved from the CDW of the Catholic Medical Center (CMC) on the use of antipsychotics in the geriatric population. All eight centers enrolled share a common electronic health database. The CDW, which incorporates seven affiliated hospitals under the CMC in Korea, is a data platform that collects and distributes clinical data to researchers, including more than 15 million electronic medical records that were completely anonymized with access granted only for research purposes following a comprehensive ethical board review [78]. We collected data on patients who were hospitalized with a first stroke diagnosis (ICD-10 codes I60, I61, I63, I64) at five CMC-affiliated hospitals in Korea between January 2011 and December 2020, and whose ages were 18 years or older. In order to extract those with de novo infection after onset of stroke, those with any preceding record of infection or admission 6 months prior to stroke onset were excluded. Before we collected the data, this study obtained approval from the Institutional Review Board of the Catholic Medical Center (HC22WIDI0003).

The exclusion criteria were as follows: subjects who were (1) not hospitalized at the time of stroke diagnosis, (2) hospitalized in departments other than neurology, neurosurgery, and rehabilitation medicine, (3) taking any immunomodulatory drugs and chemotherapy agents prior to or during stroke recovery with the exception of corticosteroids, (4) hospitalized for less than 48 h due to discharge or death, (5) infected within 2 days of admission, or (6) missing data corresponding to any variables included in the analysis.

### 4.2. Identification of Post-Stroke Infection

Post-stroke infections were identified through data extracted from the CDW. Due to the low accuracy of ICD-10 codes in identifying infections [79], laboratory results and antibiotic prescription information were used as indicators [80]. Patients with a white blood cell count above 10.0 × 10^9^/L after stroke onset who received intravenous antibiotic treatment were considered to have a post-stroke infection. The date of infection was defined as the date of the first recorded leukocytosis during the antibiotic administration period. Infections occurring within 2 days of stroke onset were excluded, as they were unlikely to be related to post-stroke infections.

Cases of post-stroke infection were categorized on the basis of the results of culture tests performed on respiratory and genitourinary samples. For respiratory samples, we classified the cases with known pathogens of hospital-acquired pneumonia [81] as positive, named “pathogen isolated from sputum”. For genitourinary samples, we classified cases with bacteria grown at densities above 10^5^ CFU as positive, named “bacteriuria”. Cases in which two or more positive pathogens from respiratory, genitourinary, or blood samples tested positive were categorized as “combined”. In contrast, positive pathogens from other sources with elevated white blood cell counts with records of intravenous antibiotic treatment were categorized as “others”.

### 4.3. Clinical Characteristics and Stroke Severity Variables

To determine the risk factors for post-stroke infections, the following clinical characteristics were extracted from the CDW: age at diagnosis, sex, comorbidities (Charlson comorbidity index (CCI)), hypertension, and atrial fibrillation), and exposure to medications including steroids, acid-suppressant drugs (proton pump inhibitors and H2 blockers), opioids, benzodiazepines, anticholinergics, dopamine agonists, selective serotonin reuptake inhibitors (SSRIs), serotonin–norepinephrine reuptake inhibitors (SNRIs), aspirin, and clopidogrel, known to be related to stroke infection [35,36,43,60].

The following variables were used to assess stroke severity and post-stroke function: modified Barthel index (MBI) [82], brain surgery (craniectomy, craniotomy, mechanical thrombolysis, intracranial stenting, carotid endarterectomy, carotid artery stenting, clipping of intracranial aneurysms, coil embolization for intracranial aneurysms, and ventriculoperitoneal shunt), mechanical ventilation, and enteral tube feeding during hospitalization. The use of mechanical ventilation was determined by the ventilator prescription code in the ICU, while enteral tube feeding was defined as a prescription for an enteral diet for more than 7 days.

### 4.4. Statistical Analyses

The statistical analysis for the study was conducted using R software version 4.1.2 from the R Foundation for Statistical Computing. The data for categorical variables are presented as frequencies and percentages, while continuous variables are displayed as means and standard deviations. Comparisons between groups were performed using Student’s *t*-test or the chi-squared test as appropriate. The impact of independent variables on post-stroke infections was analyzed using univariable and multivariable logistic regression analysis, with variables having a *p*-value < 0.1 being included in the multivariable model. The predictive performance of the combined predictors was evaluated through receiver operating characteristic (ROC) curves, and results with a *p*-value < 0.05 were considered statistically significant.

## 5. Conclusions

In summary, our study findings suggest that implementing early preventive measures in the intensive care unit, where postoperative management, mechanical ventilation, and enteral nutrition are administered, may be crucial in reducing the risk of infection. Additionally, early measures aimed at improving patients’ functional levels, such as early mobilization, should also be considered. Lastly, when caring for these high-risk stroke patients prone to infection, it is essential to evaluate the balance between the potential benefits of preventing gastrointestinal bleeding with acid-suppressant drugs or administering corticosteroids and the increased risk of infection.

## Figures and Tables

**Figure 1 antibiotics-12-00740-f001:**
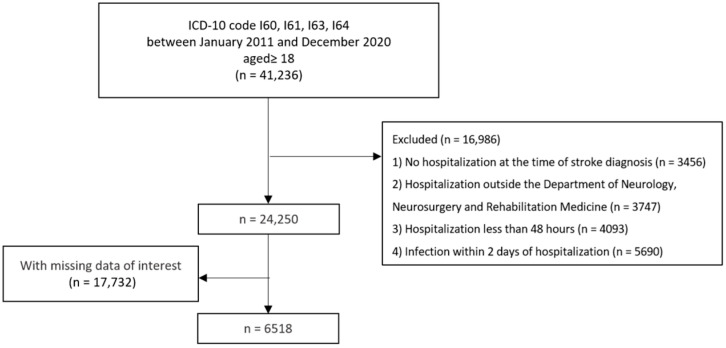
Patient flowchart. The initial screening included 41,236 patients, and 6518 patients were included in the analysis.

**Figure 2 antibiotics-12-00740-f002:**
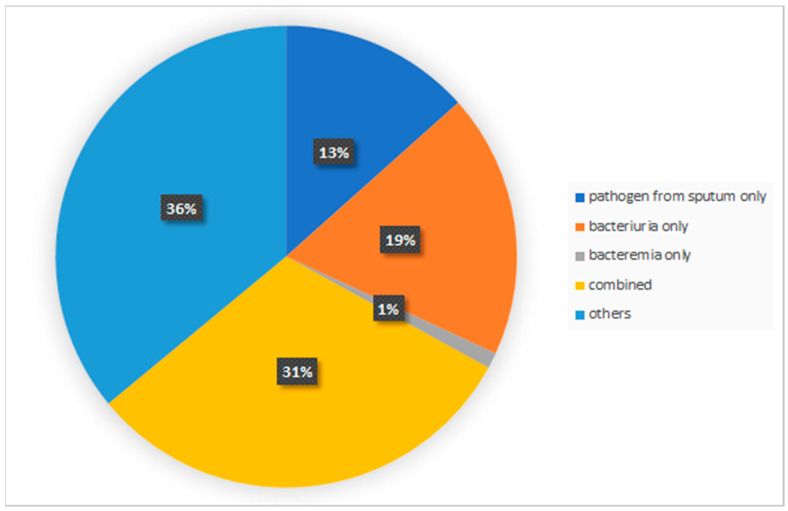
Proportions of post-stroke infection accounted for by positive culture results, where cases in which two or more positive pathogens from respiratory, genitourinary or blood samples tested positive were categorized as “combined.” In contrast, positive pathogens from other sources with elevated white blood cell counts with records of intravenous antibiotic treatment and were categorized as “others”.

**Figure 3 antibiotics-12-00740-f003:**
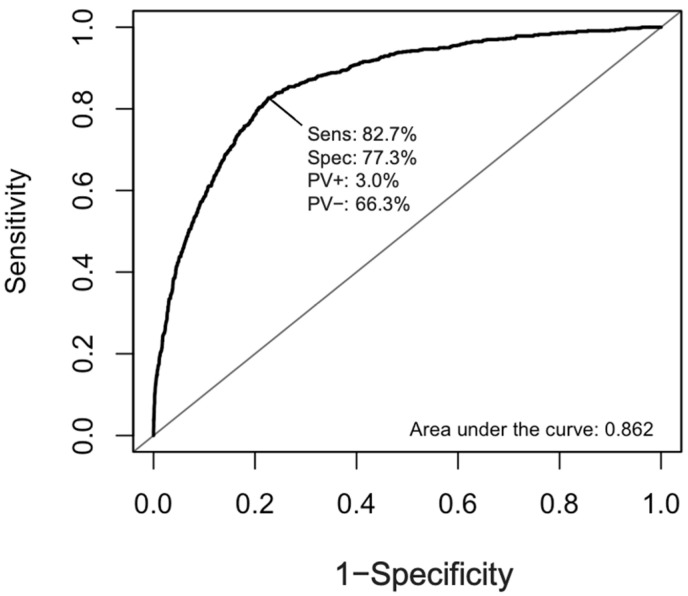
Receiver operating characteristic (ROC) curves for multivariable logistic regression models of risk factors for post-stroke infection.

**Table 1 antibiotics-12-00740-t001:** Baseline characteristics of the study population.

Clinical Variables	Total (*n* = 6518)	Infection (*n* = 797)	No Infection (*n* = 5721)	*p*-Value
Age (years), mean ± SD	67.5 ± 13.2	67.9 ± 13.6	67.4 ± 13.1	0.345
Male, *n* (%)	3736 (57.3)	499 (62.6)	3237 (56.6)	0.001 *
Hospital length of stay (days), mean ± SD	19.4 ± 75.2	38.9 ± 59.1	16.7 ± 76.8	0.001 *
Brain surgery, *n* (%)	523 (8.0)	250 (31.4)	273 (4.8)	<0.001 *
Mechanical ventilation, *n* (%)	71 (1.1)	63 (7.9)	8 (0.1)	<0.001 *
Enteral tube feeding, *n* (%)	1132 (17.4)	431 (54.1)	701 (12.3)	<0.001 *
MBI, mean ± SD	47.5 ± 31.5	23.4 ± 27.6	50.9 ± 30.6	<0.001 *
CCI, mean ± SD	3.7 ± 1.6	4.0 ± 1.7	3.7 ± 1.5	<0.001 *
Hypertension, *n* (%)	696 (10.7)	99 (12.4)	597 (10.4)	0.101
Atrial fibrillation, *n* (%)	277 (4.3)	46 (5.8)	231 (4.0)	0.029 *
Medication exposures, *n* (%)				
Antipsychotics	1194 (18.3)	212 (26.6)	982 (17.2)	<0.001 *
Opioid	111 (1.7)	22 (2.8)	89 (1.6)	0.021 *
Benzodiazepine	401 (6.2)	53 (6.6)	348 (6.1)	0.585
Anticholinergic	457 (7.0)	62 (7.8)	395 (6.9)	0.405
Dopamine agonist	349 (5.4)	88 (11.0)	261 (4.6)	<0.001 *
SSRI/SNRI	890 (13.7)	163 (20.5)	727 (12.7)	<0.001 *
Steroid	310 (4.8)	86 (10.8)	224 (3.9)	<0.001 *
Acid-suppressant drugs	4707 (72.2)	662 (83.1)	4045 (70.7)	<0.001 *
Aspirin	4753 (73.0)	484 (60.7)	4269 (74.6)	<0.001 *
Clopidogrel	4288 (65.8)	442 (55.5)	3846 (67.2)	<0.001 *

The results are presented as the mean ± SD or number (percentage). SD, standard deviation; CCI, Charlson comorbidity index; MBI, modified Barthel index; SSRI, selective serotonin reuptake inhibitor; SNRI, serotonin and norepinephrine reuptake inhibitor. * *p* < 0.05, statistically significant according to Student’s *t*-test or the chi-squared test.

**Table 2 antibiotics-12-00740-t002:** Univariable and multivariable logistic regression associated with infection.

	Univariable		Multivariable	
	OR (95% CI)	*p*-Value	OR (95% CI)	*p*-Value
Age, years				
Mean ± SD	1.00 (1.00–1.01)	0.345		
Male	1.29 (1.10–1.50)	0.001	1.79 (1.49–2.16)	<0.001 *
Hospital length of stay (days)				
Mean ± SD	1.003 (1.002–1.005)	<0.001	1.001 (1.000–1.002)	0.006 *
Brain surgery	9.12 (7.52–11.06)	<0.001	7.84 (6.24–9.87)	<0.001 *
Mechanical ventilation	61.29 (31.07–139.13)	<0.001	18.11 (8.40–44.04)	<0.001 *
Tube feeding	8.43 (7.19–9.90)	<0.001	3.61 (2.95–4.43)	<0.001 *
MBI				
Mean ± SD	0.97 (0.97–0.97)	<0.001	0.98 (0.98–0.98)	<0.001 *
Charlson comorbidity index				
Mean ± SD	1.12 (1.07–1.17)	<0.001	1.02 (0.97–1.09)	0.427
Hypertension	1.22 (0.97–1.52)	0.089	1.03 (0.78–1.37)	0.821
Atrial fibrillation	1.46 (1.04–2.00)	0.024	0.62 (0.40–0.93)	0.022 *
Other medication exposures				
Antipsychotic	1.75 (1.47–2.07)	<0.001	0.81 (0.66–1.00)	0.053
Opioid	1.80 (1.09–2.83)	0.015	1.41 (0.78–2.46)	0.235
Benzodiazepine	1.10 (0.81–1.47)	0.533		
Anticholinergic	1.14 (0.85–1.49)	0.365		
Dopamine agonist	2.60 (2.00–3.33)	<0.001	1.33 (0.97–1.82)	0.074
SSRI/SNRI	1.77 (1.46–2.13)	<0.001	1.20 (0.95–1.51)	0.124
Steroid	2.97 (2.28–3.84)	<0.001	2.18 (1.57–3.01)	<0.001 *
Acid-suppressant drugs	2.03 (1.68–2.48)	<0.001	1.43 (1.15–1.80)	0.002 *
Aspirin	0.53 (0.45–0.61)	<0.001	0.81 (0.63–1.03)	0.083
Clopidogrel	0.61 (0.52–0.71)	<0.001	0.87 (0.69–1.10)	0.241

The multivariable model included all variables with *p* < 0.1 in the univariable analysis. * *p* < 0.05, statistically significant according to multivariable logistic regression analysis. SD, standard deviation; MBI, modified Barthel index; SSRI, selective serotonin receptor inhibitor; OR, odds ratio; CI, confidence interval.

## Data Availability

The dataset is available upon request.

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
