# Peer review of "Post-Stroke Infections: Insights from Big Data Using Clinical Data Warehouse (CDW)"

_antibiotics, 2023, doi:10.3390/antibiotics12040740_

Round 1

Reviewer 1 Report

1. How long has it been since the patient got the CVA and the subsequent infection?

2. Is the infection a community-acquired infection or a nosocomial infection?

3. How about the patients’ previous 90-day history of receiving IV antibiotics? 

4.How about the patients’ previous 90-day hospitalization history? 

5.How is the GCS score related to the risk of infection? 

6. How are the comorbidities related to the risk of infection? 

7. How about other medications related to the risk of infection? 

8.Are the mucolytic agents with protective effects against pneumonia? 

9.Are the inhalation bronchodilators protective against pneumonia? 

10. How about other immunosuppressants except steroids? 

11. How about are patients with foley catheters at increased risk of an UTI? 

12. Could you please explore whether the pathogen was related to the patients’ clinical outcome? 

13.How long has it been since the patient underwent brain surgery and got an infection afterward?

14. Are the patients with oral intake at high or low risk of infection?

Reviewer 2 Report

1.     The title is interesting. The abstract is well constructed and informative.

2.     Background section is long as it need and well informative.

3.     Material and methods section is concise.

4.     Result section is informative. Quality of figures and tables is ok. They are understand by readers.

I suggest to accept this paper

Reviewer 3 Report

This paper uses available big data from Clinical Data Warehouse to perform analysis in several hospitals in Korea to understand the risk factors for hospitalized patients after acute stroke. Statistical analysis was performed in order to identify a model to predict the risk of infection during the post-stroke period, and the authors suggested that based on these findings, preventive measures and rehabilitation can be provided accordingly. Overall my recommendation is a major revision, as I have some reservations about the definition of some of the terms used related to infection. Here are several comments that the authors should consider and address during the revision of the paper:

Major comments:

1. Limitations of the authors' definition of infection should be clearly stated in the discussion part. The authors used both white blood cell count and receiving intravenous antibiotics as the definition of infection, but it should be understood that not all patients with an infection will have leukocytosis, especially in the elderly (patients with poor bone marrow reserve and response to infection), immunocompromised hosts (hematological malignancy with neutropenia), and severe sepsis (presenting with pancytopenia instead of neutrophilia). Furthermore, patients with a larger stroke volume and more severe stroke deficits have a high WBC count in the acute phase after stroke. Therefore, the authors should elaborate on this limitation in their discussion.

2. Please define a "positive test" in respiratory samples and genitourinary samples. I understand this is a retrospective study, but the authors should clearly define whether it includes bacterial culture, bacterial/ viral PCR (such as commercial platform), fungal culture, AFB smear, and culture, etc. Furthermore, as respiratory and genitourinary specimens are non-sterile specimens, different microbiology laboratory reports these specimens differently. If possible, the authors should elaborate on this.

- e.g. for urinary specimens, what is the cfu/mL that is considered as "positive" by your group or the laboratory;

- e.g. for respiratory specimens, which bacteria/pathogen would your group or the laboratory consider as "positive"  positive (Streptococcus pneumoniae is likely pathogenic, Staphylococcus aureus can be pathogenic or colonization, Candida albicans likely represent colonization).

3. Please consider using alternative terms to describe patients with infection and positive sputum culture instead of "pneumonia". Pneumonia in general is a term to describe a clinical syndrome in which the patient develops respiratory symptoms, radiologically showing pulmonary infiltrates, evidence of systemic inflammatory response, +/- a positive microbiological culture from respiratory specimens. In this study, as CXR was not analyzed for individual patients, the term pneumonia may not be appropriate in this case.

4. What are the respective lengths of stay of patients in the infection group and non-infection group, and are there any statistically significant differences? Authors may need to consider whether this is a confounder during the subsequent analysis.

Minor comments:

1. English should be improved in the manuscript in terms of choice of words (e.g. Charlson Comorbidity Index instead of Charlson Comorbidity Index score (as the term index itself already has the meaning of score), meta-analysis instead of meta-study).

2. Please double-check Table 1. The row Acid suppressive drugs "n(%)" is not necessary.

3. Review the necessity of Figure 3, as Figure 3 is just a graphical representation of Table 2.

4. In the discussion, the authors mentioned previous studies have stated atrial fibrillation is associated with post-stroke infection. Is there a possibility that it is the infection causing atrial fibrillation instead of atrial fibrillation leading to an increased risk of stroke?

Round 2

Reviewer 1 Report

Point 1. How long has it been since the patient got the CVA and the subsequent infection?

The time between a stroke onset and subsequent infection was median of 9 days (IQR: 5–14 days). I do not think that short term early rehabilitation in ICU can reduce the risk of post-stroke infection.

Point 2. How about the patients’ previous 90-day history of receiving IV antibiotics?

Response 3:In this study, we obtained hospitalization data following stroke onset. As a result,

patients' previous history before stroke for receiving IV antibiotics were not included.

This is very important information related to nosocomial infection and you must collect the data. If the database you use does not have this information, then the analysis of this database will not be applicable to clinical practice.

Point 3. How about the patients’ previous 90-day hospitalization history?

Response 4: As previously stated, data regarding previous hospitalizations were not available

through electronic chart retrieval in this study. In cases where patients were hospitalized at other

medical centers, such information would not be accessible. Nonetheless, it is crucial to acknowledge that a history of hospitalization, especially within the preceding 90 days, is a well-known risk factor for infection.

If the database you use does not have this information, then the analysis of this database will not be applicable to clinical practice.

Point 4. How is the GCS score related to the risk of infection?

Response 5: Glasgow Coma Scale (GCS) scores were not available in the Clinical Data Warehouse from which we extracted data for this study. Consequently, GCS scores were not included in our analysis.

If the database you use does not have this information, then the analysis of this database will not be applicable to clinical practice.

Point 5. How are the comorbidities related to the risk of infection?

Response 6:We appreciate your thoughtful key questions about comorbidities. For comorbidities,

previous studies have included some individual medical problems in the analysis, with a focus

mostly on atrial fibrillation and diabetes.

I do not think that patients with chronic pulmonary disease comorbidity is not a high risk of pneumonia.

Point 6. Are the mucolytic agents with protective effects against pneumonia?

Mucolytic agents such as N-acetylcysteine have antioxidant and immunomodulatory

effects, which are effective in preventing the exacerbation of chronic obstructive pulmonary disease and the development of influenza. However its protective effect in preventing post-stroke infection hand not been clearly reported and thus not included in our analysis.

(N-acetylcysteine agents have antioxidant and immuno-modulatory effects, which are effective

in preventing the exacerbation of chronic obstructive pulmonary disease and the development of

influenza [56,57].)

I do not think that N-acetylcysteine agents are effective in preventing the exacerbation of chronic obstructive pulmonary disease.

Please read following references:

1.Decramer M, Rutten-van Mölken M, Richard Dekhuijzen PN, et al. Effects of N-acetylcysteine

on outcomes in chronic obstructive pulmonary disease (Bronchitis Randomized on NAC

Cost-Utility Study, BRONCUS): a randomized placebo-controlled trial. Lancet 2005; 365:

1552–1560.

2. Zheng JP, Wen FQ, Bai CX, et al. PANTHEON study group. Twice daily N-acetylcysteine 600 mg for exacerbations of chronic obstructive pulmonary disease (PANTHEON): a

randomised, double-blind placebo-controlled trial. Lancet Respir Med 2014; 2: 187–194.

3. Cazzola M, Calzetta L, Page C, et al. Influence of N-acetylcysteine on chronic bronchitis or COPD exacerbations: a meta-analysis. Eur Respir Rev 2015; 24: 451–461.

4. Ther Adv Respir Dis 2023, Vol. 17: 1–12. DOI: 10.1177/17534666231158563

Point 7. How about other immunosuppressants except steroids?

Response 10: There are several types of immunosuppressive drugs that involve an overactive

immune system such as calcineurin inhibitors, monoclonal antibodies, immunosuppressive biologics and JAK inhibitors. These drugs increased the risk of infection by suppressing the immune system. However as stated in lines 98-99 on page 2-3, those with concomitant immunomodulatory drugs and chemotherapy agents were excluded.

You cannot exclude immunomodulatory drugs and chemotherapy agents and you should explore the issue.

Point 8. How about are patients with foley catheters at increased risk of an UTI?

The foley catheter provides a direct pathway for bacteria to enter the bladder,

increasing the risk of infection. However, the dataset in this study does not include information of foley catheters. Further research is needed.

If the database you use does not have this information, then the analysis of this database will not be applicable to clinical practice.

Point 9. Could you please explore whether the pathogen was related to the patients’ clinical

outcome?

This study did not take into account the pathogenic strains or bacterial density. Future studies should investigate whether the specific pathogen is associated with the patients' clinical outcomes.

You cannot wait for future studies to investigate whether the specific pathogen is associated with the patients' clinical outcomes. You should explore the issue.

Point 10. How long has it been since the patient underwent brain surgery and got an infection

afterward?

Why post brain surgery patients were a risk of infection?

Major comment:

When you revise your article, you have to collect more information and produce more detailed results to improve the quality and evidence of your article without doing a lot of interpretation.

Reviewer 3 Report

Thank you for revising the manuscript according to the comments. The manuscript has improved a lot, and most of my previous comments have been adequately addressed. However, I still have to recommend major revision for this manuscript as some significant errors are spotted during my review:

1. Major comments 2 and 3 (Line 113 - 115), and Figure 2:

- It would be better for authors to familiarize themselves with how the microbiology laboratory issue reports in order to use terms more accurately in the manuscript.

a) In general, for processing of respiratory specimens in the microbiology laboratory, technicians screen for pathogenic bacteria on agar plates as sputum specimens are commonly colonized by oropharyngeal flora. Therefore, it is better for the authors to use the term "pathogenic bacteria isolated from sputum" rather than "bacterial sputum" in your manuscript (as it is very common for bacteria to be isolated from sputum, just whether the bacteria is pathogenic or not).

b) Bacteruria should be defined with a cutoff using the unit of cfu/mL (colonies forming unit per ml). It is almost not possible for the microbiology laboratory to not consider bacterial density in processing urinary specimens. In general, the technician will estimate the amount of bacteria colonies on the agar plate and proceed with bacteria identification once it is above a certain cutoff, and issue "insignificance growth" if it is below a certain cutoff. It will be extremely tedious for the laboratory to identify all bacteria regardless of the cutoff. Therefore, the modified methodology from the authors is not practical, and I do not believe it is the case in your hospital.

- Therefore, I would recommend the authors discuss with the respective or involved microbiology laboratories their standard operating procedures as well as their usual practice in issuing reports, so as to refine your methodology part in this manuscript.

2. Major comment 4 (Line 223 - 226): I have noticed the authors commented that the odds ratio of 1.001 (95% CI 1.000 - 1.002), with a p-value of 0.006 is statistically not significant. However, based on the authors' methodology, a p-value of less than 0.05 is considered statistically significant. Please clarify.

Round 3

Reviewer 1 Report

No comments

Reviewer 3 Report

The authors have addressed my previous questions raised. I am impressed with the answers provided by the authors, and they have expressed their understanding of how the microbiology laboratory provided services in diagnosing the infections mentioned in the manuscript. I would recommend accepting this paper for publication.